# Neurocognitive Impairment After COVID-19: Mechanisms, Phenotypes, and Links to Alzheimer’s Disease

**DOI:** 10.3390/brainsci15060564

**Published:** 2025-05-25

**Authors:** Triantafyllos Doskas, George D. Vavougios, Constantinos Kormas, Christos Kokkotis, Dimitrios Tsiptsios, Kanellos C. Spiliopoulos, Anna Tsiakiri, Foteini Christidi, Tamara Aravidou, Liberis Dekavallas, Dimitrios Kazis, Efthimios Dardiotis, Konstantinos Vadikolias

**Affiliations:** 1Department of Neurology, Athens Naval Hospital, 115 21 Athens, Greece; doskastr@gmail.com (T.D.); dantevavougios@hotmail.com (G.D.V.); konkormas@med.uoa.gr (C.K.); k.spiliopoulos@ac.upatras.gr (K.C.S.); tamara.arav@gmail.com (T.A.); dekavallas.liberis@gmail.com (L.D.); 2Department of Neurology, General University Hospital of Alexandroupoli, 681 00 Alexandroupoli, Greece; anniw_3@hotmail.com (A.T.); christidi.f.a@gmail.com (F.C.); vadikosm@yahoo.com (K.V.); 3Department of Neurology, Faculty of Medicine, University of Cyprus, 1678 Lefkosia, Cyprus; 4Department of Respiratory Medicine, Faculty of Medicine, School of Health Sciences, University of Thessaly, 413 34 Larissa, Greece; 5Department of Physical Education and Sport Science, Democritus University of Thrace, 691 00 Komotini, Greece; ckokkoti@affil.duth.gr; 6Third Department of Neurology, Aristotle University of Thessaloniki, 541 24 Thessaloniki, Greece; dimitrios.kazis@gmail.com; 7Department of Neurology, Faculty of Medicine, School of Health Sciences, University of Thessaly, 382 21 Larissa, Greece; edar@med.uth.gr

**Keywords:** COVID-19, cognitive impairment, long COVID, neurodegenerative disease

## Abstract

Background/Objectives: SARS-CoV-2 can affect the central nervous system directly or indirectly. AD shares several similarities with long COVID cognitive impairment on a molecular and imaging level, as well as common risk factors. The objective of this review is to evaluate the incidence of post-acute COVID-19 cognitive impairment. Secondarily, we aim to determine if neuroinflammation in COVID-19 survivors may be associated with the onset of neurological disease, with a focus on Alzheimer’s disease (AD). Methods: literature search up to March 2025 on the prevalence of cognitive deficits in COVID-19 survivors, underlying pathophysiology and associations with neurological disorders. Results: a wide array of neuropsychiatric manifestations is associated with COVID-19; executive function, memory, and attention are the most frequently reported neurocognitive deficits, regardless of COVID-19 severity. There are associations between the risks for cognitive deficits post-infection with the age of the patients and the severity of the disease. Increasing evidence suggests that neurocognitive deficits are associated with the onset of neurological and neuropsychiatric disease in COVID-19 survivors. Conclusions: clinicians caring for COVID-19 survivors should actively investigate neurocognitive sequelae, particularly for patients with increased risk for cognitive deficits.

## 1. Introduction

Although COVID-19 was initially defined as a novel pneumonia caused by severe acute respiratory syndrome coronavirus type 2 (SARS-CoV-2) [1], multisystem involvement both during COVID-19 infection and following the resolution of its acute phase are well-established features of the disease [2]. It has been estimated that approximately 20–60% of COVID-19 survivors suffer from prolonged persistent effects after acute COVID-19 [1,3]. The term “post-acute sequelae of SARS-CoV-2 (PASC)” or long COVID (LC) has been used to collectively describe the manifestations that develop during or after an infection consistent with COVID-19 and continue for more than 12 weeks and are not explained by an alternative diagnosis, according to the National Institute for Health and Care Excellence (NICE) guidelines [1,3,4,5]. Thus, PASC is primarily characterized by constitutional symptoms, such as fatigue, loss of smell, malaise, dyspnea, and cough [1,3,4,5], but affects multiple organ systems and functions, including cerebrovascular, neurological, neuropsychiatric, and speech and language [1,2,3,4,5]. Additional manifestations include cardiovascular symptoms, acute kidney injury, gastrointestinal symptoms, liver injury, hyperglycemia and ketosis, neurological complications, ocular symptoms, dermatologic complications, and a variety of neuropsychiatric syndromes [1,2,3,4,5]. It is estimated that COVID-19 can cause a broad spectrum of chronic non-specific neurological or mental complications, including neurocognitive impairment, even with a mild disease course, in over one-third of patients [6,7]. Neurological complications, other than fatigue, include dizziness, myalgia, sleep problems, migraine, headaches, dysgeusia, ageusia, hyposmia, anosmia, neurocognitive impairment, encephalopathy, encephalitis, disorientation, poorly organized movements to commands, Guillain-Barre syndrome, and stroke [4,7,8,9,10,11].

The occurrence of neurological complications as a response to underlying viral infections, regardless of the pathogen, is documented [12,13,14,15]. Increasing evidence suggests that a causal relation between COVID-19 and neurological/mental disorders, such as psychosis, may exist [16]. New onset of neurocognitive deficits can appear in the acute phase of the disease, at recovery or with delayed onset up to six months post-infection [6,7,9,17,18,19]. It is estimated that approximately up to one-third of COVID-19 survivors suffered from at least 1 neurological complication [11]. The Nautilus project, which was launched in June 2021, demonstrated that COVID-19 patients had worse performance in global cognition as well as individual neurocognitive tasks (namely learning and long-term memory, processing speed, language and executive functions) compared to healthy controls on neurocognitive tests that are used in routine clinical practice [10,20]. Overall, these symptoms can affect functional independence, everyday activities, and cognitively demanding functions, such as work performance, employment, and overall quality of life, even in COVID-19 patients with mild cognitive impairment, as soon as 12 weeks post-diagnosis [13,17,21]. Deficits in executive functioning and neuropsychiatric symptoms (anxiety and depression) correlate with poor quality of life in severe COVID-19 patients as soon as 90–120 days from hospital discharge [21]. Poor concentration was also the most frequently reported cognitive deficit that interfered with poor quality of life assessments up to 12 months post-diagnosis [10]. It is, therefore, of vital importance to identify the cognitive sequelae of COVID-19 promptly and manage them appropriately. Taken together, the superimposition of mechanisms over phenotypes indicates that a deeper, potentially causative connection may exist between COVID-19 and impaired cognition [13].

The association between systemic viral infections and the onset of neurodegenerative diseases, including dementia, is also well-established, though the mechanisms have not yet been fully elucidated [22,23,24,25,26]. Omics approaches (transcriptomics and proteomics) have identified associations between COVID-19 and Alzheimer’s disease (AD) at transcription level as well as between infection-relevant proteins, cognitive deterioration, and brain volume loss, mainly in the temporal lobe, including plasma biomarkers which are hallmarks of AD dementia, such as phosphorylated tau-181 (p-Tau-181) and amyloid beta (Aβ42, Aβ42:40 ratio) [8,13,27,28], confirming the cross-talk between SARS-CoV-2 infection and the long-term risk of dementia [26]. Notably, lower Aβ42 and Aβ42:40 levels and increased p-tau-181 CSF and plasma levels are detected in the presymptomatic stage of AD, though the changes in the plasma were less dynamic compared to the CSF [29,30]. A two-sample Mendelian randomization analysis based on genome-wide association studies of susceptibility, hospitalization, and severity of COVID-19, and six major neurodegenerative disorders, namely Alzheimer’s disease (AD), amyotrophic lateral sclerosis, frontotemporal dementia, Lewy body dementia, multiple sclerosis, and Parkinson’s disease (PD), demonstrated a statistically significant positive genetic correlation between COVID-19 hospitalization and AD only onset as well as between hospitalization and severity of COVID-19 with a higher future risk for AD [31]. It is therefore hypothesized that there may be an association between SARS-CoV-2 infection and AD onset, which has been confirmed at biomarker level too and is subject to modulation by several factors including age [32].

The objective of the present study is to perform a critical evaluation of up-to-date evidence on the mechanisms and phenotypes of cognitive impairment in COVID-19 survivors and its potential association with the onset of neurological disease, with a focus on AD.

## 2. Materials and Methods

A thorough search of the literature on PubMed/Medline was performed for articles published up to 15 March 2025 using the following keywords: (cognitive or neurocognitive impairment or cognitive deficits), (COVID-19 or SARS-CoV-2 or long-COVID), (prevalence or epidemiology), cognitive phenotypes of COVID-19, biomarkers of cognitive impairment and Alzheimer’s disease. Selected admissible publications were then searched for further pertinent references. Evidence on the epidemiology of neurocognitive impairment was derived from systematic reviews and meta-analyses because they contributed greater validity compared to individual studies. Other publications (such as case series, cohort studies) were only used when their findings could contribute important information. However, it is acknowledged that the conclusions from these studies should be treated with caution, due to the inherent limitations, including small sample, lack of suitable controls for comparison purposes, or limited information on confounders that could not allow objective assessments to be made.

## 3. Candidate Mechanisms of Neurocognitive Impairment Due to SARS-CoV-2 Infection

SARS-CoV-2 was identified in autopsies of COVID-19 patients throughout the entorhinal cortex and hypothalamus, indicating both viral entry and dissemination [33]. The initially predicted, at the onset of the pandemic, neurotropic potential of human coronavirus is well-established and confirmed via elevated levels of neuronal damage markers in COVID-19 survivors [34]. Other than the hippocampus, the preferred sites for injury following SARS-CoV-2 infection include the temporal and frontal lobe regions that map to specified cognitive abilities [9,35]. Cerebellar and prefrontal lesions account for the deficits in executive functioning and memory of patients suffering from LC [21,36,37]. Neuropathological evidence combined with neuroimaging studies has indicated that SARS-CoV-2 can enter into the central nervous system (CNS) via the transcribial route and migrate trans-synaptically via the olfactory-hippocampal projections [35]. In the meta-analysis by Kim et al., olfactory bulb abnormalities were the most frequently observed neuroimaging finding in COVID-19 patients, of whom only 3.4% had relevant findings on neuroimaging examination, with critically ill patients having more findings than other patients (9.1% vs. 1.6%; *p* = 0.029) [38].

SARS-CoV-2 may use the proximity of the olfactory mucosal, endothelium, and neurological tissue, including sensitive olfactory and sensory nerve endings, via binding to angiotensin-converting enzyme (ACE2) receptors, to reach the nervous system through the olfactory mucosa’s neural–mucosal interface [33,39]. Although ACE2 expression is low in the CNS, upregulation of ACE2 expression is observed in AD patients and is positively correlated to the severity of the disease [40]. The upregulation of ACE2 expression in AD patients may make them more prone to SARS-CoV-2 infection and accelerate neuronal damage [27,41,42]. Furthermore, alternative means of entry via hijacking other host factors, such as antiviral membrane proteins, are also proposed [43].

Direct neuroinvasion is not the sole mechanism of neuronal damage and subsequently neurocognitive deficits. There are further factors that contribute towards neuroinflammation and neuronal damage that are collectively presented with neurocognitive impairment [27]. These include systemic inflammation, peripheral tissue damage, oxidative stress, innate immune system activation, prolonged antigenic stimulation, and cytolytic β-cell damage [27,44]. Growing evidence suggests that peripheral immune cells, including autoreactive T cells, play a critical role in maintaining CNS balance [45]. Long-term peripheral inflammation can cause microglia activation and overall dysregulation of immune activity within the CNS that can initiate or exacerbate neurological damage [46,47,48]. Experimental and clinical findings have shown that systemic infections—viral or bacterial—can modulate brain function both through neural connections and circulating inflammatory molecules [23,46,47,49,50]. Moreover, immune cell activation in the periphery and subsequent migration in the CNS can cause stimulation of resident microglia, which fosters neuroinflammation. For example, Herpes simplex virus (HSV) can causes chronic inflammation that promotes Aβ overproduction and aggregation [15]. Additionally, growing research has highlighted the connection between gut microbiota composition and neurodegenerative disorders. Disruptions in gut microbial balance disruption can lead to intestinal inflammation and potentially evolve to systemic inflammation through immune-stimulating factors and microbial metabolites entering the bloodstream [47,51]. In addition to indirect effects, gut bacteria can mitigate neuroinflammation directly via interaction with receptors found on vagus nerve fibers [47,52].

Using omics approaches in patients with LC versus non-LC eight months post-infection, altered distribution of immune cell subsets as well as higher levels of SARS-CoV-2 antibodies in LC patients were observed, which were indicative of immune dysregulation/hyperactivation and systemic inflammation [53]. Long COVID patients with neurocognitive and neuropsychiatric symptoms have a persistently distinct cytokine profile compared to uninfected controls, which reflects a distinct immune cell repertoire consistent with the neurotoxic effect of cytokines in the CNS [54,55,56,57]. Further evidence in favor of this is provided by studies of genes that are differentially expressed in AD and COVID-19 patients and are associated with immune responses and cytokine storm [41]. Abnormal oligoclonal banding (OCB) in the CSF samples of COVID-19 patients versus controls that most likely reflected systemic autoantibody production has also been reported [6,9]. Furthermore, type I interferon signaling dysregulation, a hallmark of both viral infections (including COVID-19) and AD, provides yet another shared mechanism that may even directly mediate cognitive effects [58]. Transcriptomic studies in COVID-19 brains demonstrate that severe COVID-19 infection is linked with molecular patterns that are consistent with the aging of the human brain [59].

There are additional factors that contribute to neuronal damage in COVID-19 survivors. Thus, hypoxia compounded by potential intubation can mitigate a highly inflammatory environment that can aggravate neuronal cell injury and inflammation. The occurrence of seizures, which is common in approximately 2.2% of COVID-19 patients regardless of pre-existing epilepsy, though higher in patients with pre-existing epilepsy, is also a risk factor [60]. Altogether the aforementioned confer an increased risk and/or occurrence of complications, manifesting as encephalopathies, ischemic infarctions, white matter damage, and coagulopathies [21,38,61] resulting in cerebrovascular disease that contributes further to neurocognitive deterioration [8,13,27,57]. Cortical atrophy, hypoxia, and cerebrovascular or vascular disorders, such as atherosclerosis, usually develop secondarily to neuroinflammation and may thus account for the persistence of neurocognitive symptoms in studies involving long-term follow-up [7,62]. Notably, hypoxia on its own may lead to cognitive impairment regardless of the cause [63].

The effect of confounding factors has to be taken into account [12]. Neuroinflammation due to SARS-CoV-2 infection can be aggravated by age-related changes that predispose to higher immune system frailty [13,64]. COVID-19 severity is associated with age, which is an established risk factor for neurocognitive impairment in the general population [12]. Mental health disorders, which may either pre-exist or emerge as complications of COVID-19, as well as the psychosocial strain of COVID-19 due to social distancing place an additive burden on cognitive performance [65]. Additional confounders include cerebral oxygenation, the effect of mechanical ventilation, intubation, and the occurrence of acute respiratory distress syndrome (ARDS) [12,66]. It is suggested that invasive intervention (i.e., mechanical ventilation) may be associated with fewer neurocognitive sequelae (in visuospatial/executive functions, naming, short- and long-term memory, abstraction, and orientation) compared to oxygen supply in hospitalized COVID-19 patients [67]. Neurocognitive reserve, which may be indirectly evaluated from years of education, may exert a potentially protective effect on severe COVID-19-related neurocognitive impairment by modulating the impact of ICU-relevant variables, such as delirium and mechanical ventilation, on neurocognitive functioning [66]. Delirium, an acute reversible neurocognitive deficit that affects more than 50% of ICU-admitted patients, mainly elderly [12,66], is commonly reported in critically ill COVID-19 patients [68]. Furthermore, ARDS, which alone is a risk factor for neurocognitive impairment and delirium, is commonly observed in COVID-19 patients [12,21,69,70]. The vast majority of patients who develop ARDS, regardless of COVID-19, have neurocognitive impairment already at hospital discharge (70% up to 100% of patients), which can be sustained up to 5 years later (in 20% of patients) [71]. The importance of insulin regulation in cognitive performance overall and, particularly, learning and memory is established [72]. Diabetes and hypertension, which can increase the risk of stroke, vascular dementia as well as a history of attention deficit hyperactivity disorder may also make COVID-19 patients more vulnerable to executive functioning deficits [6]. Notably, uncontrolled hypertension has been associated with COVID-19 severity [73] as well as increased tau (total and p-tau-181) CSF levels in cognitively healthy subjects, but not Aβ42 levels, [74] which highlights the potential role of vascular injury in AD onset [32]. The role of sex has also been investigated with inconclusive evidence. There is inconclusive evidence on the effect of sex in COVID-19 severity and severity-modulating factors such as hypertension [73,75].

## 4. Overview of COVID-19 Neurocognitive Sequelae (Table 1)

COVID-19 may aggravate pre-existing cognitive deficits that might have been undetected, and in approximately 22% of COVID-19 survivors, neurocognitive deficits develop de novo [17]. However, the neurocognitive assessment of COVID-19 patients is challenging. Cognitive impairment following SARS-CoV-2 infection can be subclinical, despite typical laboratory and imaging results, and therefore not diagnosed [76,77]. Longitudinal studies of up to 10 months follow-up on LC patients with new onset neurocognitive deficits report that the neurocognitive deficits may resolve over time, albeit in a minority of patients [18]. Alternatively, neurocognitive deficits may not potentially be diagnosed promptly in critically ill COVID-19 patients due to decreased levels of consciousness [38]. The majority of published evidence on neurocognitive deficits involves patients with severe COVID-19, potentially because patients with mild to moderate disease were home-restrained.

**Table 1 brainsci-15-00564-t001:** Meta-analyses that have contributed information on neurocognitive impairment in COVID-19 patients.

Author, Year	Study Design	Primary Study Objective	Sample Type and Evaluation Date	Sample Size	Participant Age	Cognitive Assessment Tool Used	Main Results	Additional Results from Subgroup Analysis
Tavares-Júnior et al., 2021 [8]	Systematic review	Cognitive impairment in acute vs. subacute phase COVID-19 patients (<12 weeks from COVID-19 onset vs. >12 weeks from COVID-19 onset).	A total of 16 studies of COVID-19 patients with new cognitive impairment or deteriorated from previous cognitive impairment after infection before or at 12 weeks of COVID-19 infection; 3 studies assessed cognitive impairment after 12 weeks.	1245 patients who were cognitively evaluated and 61 controls	Median age 36.2 years (SD = 11.7) to 67.23 years (SD = 12.89)	Various tools	Cognitive impairment varied from 2.6% to 81%. In studies after 12 weeks, cognitive impairment varied from 21% to 65%. The individual studies mentioned are cited below	-
Misra et al., 2021 [11]	Systematic review and meta-analysis	Determine neurological manifestations.	COVID-19 patients, of whom 89% were hospitalized.	145,721 COVID-19 patients	Not reported	Not reported	The pooled prevalence was: - Stroke 2% (95% CI 1–2%) - Neuropsychiatric disorders 24% (95% CI 2–61%) Neuropsychiatric disorders were found to affect 1 in 4 hospitalized patients	Subgroup analyses were conducted in studies that included or disaggregated data on patients with COVID-19 who were ≥60 years and those <18 years of age presenting with neurologic symptoms. For the elderly, we found 13 studies reporting solely on older patients and 2 studies with disaggregated data, representing 3176 hospitalized patients presenting with 10 neurologic symptoms, with the most prevalent (95% CI, number of studies) as follows: acute confusion/delirium 34% (95% CI 23–46%, 5 studies), fatigue 20% (95% CI 11–31%, 9 studies), myalgia 11% (95% CI 7–15%, 10 studies), dizziness 5% (95% CI 2–9%, 3 studies), and headache 5% (95% CI 2–8%, 10 studies). Acute confusion/delirium affected 1 in 3 hospitalized older patients with COVID-19 are affected (pooled prevalence 34%) compared with 5% in young adults and 12% for all ages.
Houben et al., 2022 [10]	Systematic review and meta-analysis	Relationship between COVID-19 and cognitive functions up to 1 year after hospital discharge.	Both hospitalized and non-hospitalized patients.	90,317 COVID-19 patients and 3786 control patients	53.8 (10.4) years; Level of education 12.6 (2.7) years	Various	Meta-analysis performed on 959 participants, 513 patients demonstrated that long COVID-19 patients had, on average, a decrease of −0.41 [95% CI −0.55–−0.27] (using fixed effect model due to low heterogeneity (Tau2 = 0.0047, *p* = 0.32)	
Premraj et al., 2022 [7]	Meta-analysis	Determine the prevalence of neurological and neuropsychiatric symptoms reported 12 weeks (3 months) or more after acute COVID-19 onset in adults.	Included hospitalized and non-hospitalized patients, both with and without ICU admission.	10,530 patients were evaluated	Mean (SD): 52 (10) years	Various	Overall prevalence for neurological post-COVID-19 symptoms were: fatigue (37%, 95% CI: 24–50%), brain fog (32%, 9–55%), memory issues (27%, 18–36%), attention disorder (22%, 10–34%), myalgia (18%, 4–32%), anosmia (12%, 7–17%), dysgeusia (11%, 4–17%), and headache (10%, 1–21%). Neuropsychiatric conditions included sleep disturbances (31%, 18–43%), anxiety (23%, 13–33%), and depression (12%, 7–21%).	Short (3 to 6 months) versus long-term (>6 months) follow-up indicated there was either no increase or slightly higher prevalence in neurocognitive deficits over time.
Crivelli et al., 2022 [12]	Systematic review and meta-analysis	Determine neuropsychological test performance either during the acute phase of COVID-19 or after recovery (up to 7 months post-infection).	COVID-19 patients with no previous cognitive impairment vs. comparison group (healthy controls with no history of COVID-19 infection or patients enrolled pre-pandemic). COVID-19 patients ranged from asymptomatic to severe infection that required ICU admission.	2103 patients and 506 healthy controls	Mean age of COVID cases: 56.05 years range (50.03 to 62.07) vs. controls: 50.30 years (range 43.56 to 57.05)	Various	The occurrence of cognitive impairment in the acute COVID-19 phase ranged from 61.5% (mild to moderate) to 80% (moderate to severe patients); impairment in executive functions, attention, and memory were found in post-COVID-19 patients.	Meta-analysis on a subgroup of 290 individuals showed a difference in MoCA score between post-COVID-19 patients versus controls (mean difference = −0.94, 95% confidence interval [CI] −1.59–−0.29; *p* = 0.0049).
Shan et al., 2024 [78]	Systematic review and meta-analysis	Investigate the relationship between COVID-19 infection and a possible increased likelihood of older adults (≥60 years) in developing new-onset dementia (NOD).	Median observation period 12 months post-infection. Both hospitalized and outpatients were included in the analyses.	939,824 post-COVID-19 survivors and 6,765,117 controls	Not reported	Not reported	The overall incidence of NOD was about 1.82% in the COVID-infected group compared to 0.35% in the non-COVID-infected group. The overall pooled meta-analysis showed a significantly increased NOD risk among COVID-19 older adult survivors compared to non-COVID-19 controls (RR = 1.58, 95% CI 1.21–2.08).	The risk for NOD in the COVID-group was compared to two types of control groups: non-COVID cohorts with other respiratory infections [control group (C1)]. COVID-19 group and C1 group shared a comparably increased risk of developing NOD (overall RR = 1.13, 95% CI 0.92–1.38).
Zhang et al., 2025 [79]	Systematic review and meta-anayses	Determine the risk of NOD in adult patients; patients with known dementia or lacked adequate data about the risk of dementia were excluded.	Patients were followed for up to 24 months post-infection.	26,408,378 participants were included.	Not reported	Not reported	Pooled analysis indicated COVID-19 was associated with an increased risk of new-onset dementia (HR = 1.49, 95% CI: 1.33–1.68). This risk remained elevated when compared with non-COVID cohorts (HR = 1.65, 95% CI: 1.39–1.95) and respiratory tract infection cohorts (HR = 1.29, 95% CI: 1.12–1.49), but not influenza or sepsis cohorts. Increased dementia risk was observed in both males and females, as well as in individuals older than 65 years (HR = 1.68, 95% CI: 1.48–1.90), with the risk remaining elevated for up to 24 months.	

Abbreviations HR: Hazard ratio; ICU: Intensive care unit; MoCA: Montreal Cognitive Assessment; ΝOD: New-onset dementia; RR: relative risk.

A broad range of neurocognitive impairment deficits, extending from 20% to 65% of patients, is reported for COVID-19 survivors who were followed-up from 12 weeks post-COVID-19 onset or post-hospital discharge for up to 12 months [8,17,18,19,80,81,82,83]. The use of global cognitive assessment tools in mainly hospitalized patients who were in the acute phase of COVID-19 (within the first three months) confirmed that 50% of patients presented with scores that were consistent with neurocognitive deficits [80]. Similar to the variability in the extent and severity of cognitive impairment among COVID-19 patients, the timing (onset) of neurocognitive impairment relative to COVID-19 infection can be heterogeneous, i.e., neurocognitive impairment can be detected shortly after infection or later [6,7,8,17]. In the latter case, the identification of potentially direct correlations between COVID-19 and neurocognitive impairment could be challenging because COVID-19 may aggravate risk factors that can overall contribute to neurocognitive impairment [11]. These include age as well as pre-existing neurocognitive deficits/dementia that can be modulated by the severity of COVID-19 infection [77].

Limited cross-sectional studies on cohorts that included both hospitalized and non-hospitalized patients have provided evidence on the prevalence of multiple versus single neurocognitive deficits in COVID-19 survivors [83,84]. Ferucci et al. reported that 21% of hospitalized non-ICU patients had deficits both in information processing speed and verbal memory five months after hospital discharge [83].

A review of evidence on neurocognitive deficits in LC patients is presented.

### 4.1. Long COVID

The Human Phenotype Ontology working group attempted to define the spectrum of long-COVID neurocognitive deficits and reported that the majority of deficits were related to memory, namely memory impairment, short and long-term memory impairment, procedural memory loss, and anterograde memory impairment [2]. Procedural memory is a category of long-term memory that supports the implicit learning of new information and is linked to both working memory and episodic memory [85]. Emotional and behavioral abnormalities such as depression and anxiety were also identified in LC patients [2,86]. Other than memory, neurocognitive deficits that are reported in LC involve executive skills, notably phonemic fluency and attentiveness, attention, visual-spatial ability, as well as multi-domain impairment in more than one neurocognitive ability or overall lower cognition scores [12,18]. These deficits could persist up to seven months following COVID-19 recovery [12].

Brain fog is also increasingly retrieved on publications of LC to collectively describe sleep problems, memory impairment, slow thinking, confusion, lack of concentration, difficulty in focusing, and overall decreased neurocognitive ability or mild neurocognitive impairment [87]. Albeit the inconclusive evidence, sleep contributes to learning and memory consolidation; thus, sleep problems can contribute or aggravate memory impairment in LC patients [88].

There have been some attempts to identify “phenotypes” of LC [89]. Based on the results of an online survey on 2550 non-hospitalized COVID-19 survivors who were within the first 2 weeks of infectionthat presented with early (acute) and persistent symptoms over a median duration of 7.7 months, the acute manifestations could be grouped into two clusters: a primary cluster with cardiopulmonary symptoms comprising 88% of the subjects, and a smaller cluster with non-specific multi-system symptoms. Persistent symptoms could, also, be grouped into two clusters: a small cluster with multi-system symptoms and a large cluster, comprising 88.8% of the subjects, with cardiopulmonary, neurocognitive, and exhaustion symptoms. This shift in the large persistent symptom cluster could be predicted by female gender, younger age, worse baseline health status and inadequate rest during the first 2 weeks from infection [89].

Premark et al. [7] assessed neurological and psychological symptoms of LC that persisted short-term (3–6 months) and long-term (>6 months) from COVID-19 onset by performing a meta-analysis of 10,530 patients from eligible studies with high heterogeneity. Approximately 50% of the patients were hospitalized and 13% were admitted in ICU. Fatigue (35%), anosmia (12%), brain fog (32%), memory problems (28%), attention disorder (22%), myalgia (18%), dysgeusia (11%),and headache (10%) were the most frequently reported neurological problems. Meta-analyses of short versus long-term follow-up indicate there is either no increase or a slightly higher prevalence in neurocognitive deficits over time [7,17]. There is conflicting evidence, however, on the prevalence of neuropsychiatric problems; either no increase or significantly higher prevalence is reported [7,17]. In the meta-analysis by Premraj et al., the slightly higher prevalence of neurocognitive impairment long-term could be attributed to the concomitant significantly increased prevalence of neuropsychiatric problems, which in turn affect long-term neurocognitive performance [7]. The long-term persistence of the neurocognitive deficits that are associated with LC could reflect both the underlying pathology and the increased neuropsychiatric symptom burden. As a novel condition, however, the natural history and evolution of this phenomenon remain to be determined longitudinally.

Although information on LC in children is limited compared to adults, children may experience similar long-term neurological effects of COVID-19 as adults [90]. Neurological symptoms that have been reported by parents of children affected with LC 6–8 months after disease onset included headaches, difficulties in concentrating, dizziness, memory loss, depression and, to a lesser extent, sleep disorders [90]. The children in the aforementioned publication had a mild disease course. An ongoing long-term follow-up study of child neurodevelopment that included general childhood neurocognitive scores in 2020–2021 vs. the preceding decade (2011–2019) reported that children who were born during the pandemic had significantly reduced verbal, motor, and overall neurocognitive performance compared to children born in the period prior to the pandemic [91]. This effect was more pronounced in lower socioeconomic status families [91]. This study supports that even in the absence of COVID-19 illness, the social changes alone that the pandemic conferred had an important impact on the neurocognitive status of children [91].

Based on electronic health record data of approximately 1.3 million COVID-19 patients of all ages, Taquet et al., 2022 [92] estimated the 2-year risk trajectories of 14 neurological and psychiatric diagnoses in three age groups (<18 years, adults aged 18–64 years, and older adults aged ≥ 65 years), as well as whether and when these risks returned to baseline. In the adult populations, the risk of mood and anxiety disorders returned to baseline one to two months post-infection. However, the risk of neurocognitive deficit, dementia, psychosis, and epilepsy remained elevated two years post-infection. Although children did not have an increased risk of mood or anxiety problems 6 months post-infection, they had an increased risk of neurocognitive deficit, insomnia, intracranial hemorrhage, ischemic stroke, nerve, nerve root, and plexus disorders, psychotic disorders, and epilepsy or seizures. Compared to a matched control of patients with other respiratory infections, the risk of neurocognitive deficits in children was transient with a finite risk horizon of 75 days, whereas the finite time to equal incidence was 491 days [92].

### 4.2. Neurocognitive Deficits and COVID-19 Severity

The effect of COVID-19 severity-related factors on neurocognitive functioning is increasingly investigated [12,81,93]. Neurocognitive performance has not been extensively investigated in asymptomatic COVID-19 patients, although they also suffer from neurocognitive sequelae [13,77]. Compared to healthy controls, asymptomatic COVID-19 patients of a broad age range performed statistically significantly worse in visuoperception, naming, and fluency [12,94]. A single-center study of a small cohort of patients (n = 35, age range 20–60 years old) without pre-existing mental or neurocognitive deficits demonstrated that compared to asymptomatic patients, patients with symptoms of headache, anemia, indigestion, diarrhea, and hypoxia, with a need for oxygen supply, had lower performance on memory, attention, and executive function tests, as well as lower global cognition score [95]. Neurocognitive symptoms correlated with anxiety and depression, whereas the most prominent deficits based on neurocognitive measures involved memory, attention, and semantic fluency (5.7% of patients), working memory and mental flexibility (8.6%), and phonetic fluency (11.4%) [95].

COVID-19 patients with mild to moderate disease performed worse in global cognition assessments compared to healthy controls or pre-pandemic neurocognitive assessments at a follow-up of 85 days up to 7 months post-infection [12,19,76]. The meta-analysis by Crivelli et al. that included patients across various severity levels (mild to severe) who were in the acute COVID-19 phase demonstrated that neurocognitive impairment ranged from 61.5% (mild-to-moderate patients who were hospitalized in general hospitals) up to 80% (moderate-to-severe patients who were admitted in a rehabilitation clinic), of whom 40% also presented with mild to moderate depression [12].

Although individual studies may report associations of various strengths between COVID-19 severity and neurocognitive impairment, increasing evidence suggests that the severity of COVID-19 may be concordant with the severity of cognitive deficits; thus, hospitalized patients could be more severely affected [12,39,86]. It is suggested that at least 30% and approximately up to 67% of hospitalized patients with severe COVID-19 experience neurocognitive impairment [9,21,96]. The most frequently reported neurocognitive deficits in severe COVID-19 include disorientation, confusion, disorganized movements, and inattention, suggesting executive dysfunction or dysexecutive syndrome, delirium, encephalopathy, disorientation, encephalitis, psychosis, and difficulties in verbal fluency, processing speed, encoding, episodic, working memory, and learning [9,12,21,66,93,96,97,98]. Delirium is probably the most frequently reported neurocognitive deficit in hospitalized patients with severe COVID-19 [96].

Increasing evidence accumulates from hospitalized ICU-admitted patients. Up to 2.6% of invasively ventilated COVID-19 survivors with ARDS and up to 81% of hospitalized patients suffer from neurocognitive deficits [8]. The neurocognitive deficits that were more frequently reported > 12 weeks post-infection in the aforementioned pooled analysis included information processing speed, memory, executive function, verbal recall, naming, visuoperception, fluency, and concentration, whereas language deficits were rarely reported [8]. In the prospective study of Ollila et al. on neurocognitive deficits in ICU-treated COVID-19 patients with more than 12 years of education, attention appeared to be most affected overall, regardless of gender and executive functions in men [99]. ICU patients may suffer from greater extent and severity of neurocognitive impairment and neurological deficits compared to non-hospitalized patients (home-restrained) or patients requiring less extensive treatment or non-COVID controls [12,99,100,101]. Furthermore, a correlation between the length of time spent in the ICU and the low global cognitive functioning score has been reported [93], although the evidence is conflicting [81]. However, both global cognitive impairment and deficits in executive function were associated with the severity of respiratory symptoms and poorer pulmonary function in ICU-treated patients [81].

It is suggested that the neurological sequelae of severe COVID-19 include an amnesic and dysexecutive syndrome with attention deficits that may coexist with neuropsychiatric manifestations such as depression and anxiety [12,21]. It was not, however, clear if the deficits detected could persist long-term or could trigger or accelerate the onset of neurodegenerative diseases [21].

In addition to neurocognitive deficits, COVID-19 disease severity correlates with the prevalence of psychiatric symptoms and other aggravating risk factors for neurocognitive impairment [39]. In a pooled analysis of observational studies that included overall 145,721 COVID-19 patients, mainly hospitalized (89% of patients) neuropsychiatric disorders were found to affect 1 in 4 hospitalized patients [11]. Stroke was the most frequently reported neurologic diagnosis overall (pooled prevalence 2%) at 5% in young adults and 12% across all ages, whereas acute confusion/delirium was prevalent in 34% of patients aged ≥ 60 years compared with 5% in young adults and 12% for all ages [11]. Among hospitalized patients, the age of 59 years has been reported as the threshold at which COVID-19 patients were more prone to exhibit neurocognitive deficits [80]. A retrospective analysis of electronic health records of >236,000 COVID-19 survivors in the first six months after infection reported an estimated incidence of first-time neurologic or psychiatric diagnosis of 12.84% (12.36–13.33) in the overall cohort, which was 2-fold higher for ITU-admitted patients [25.79% (23.50–28.25)] [86]. Additionally, ITU-admitted patients had a 2–3-fold higher risk for the development of neurodegenerative conditions (parkinsonism and dementia), namely anxiety disorders and psychotic conditions [86]. Additionally, an important finding that has emerged from the available literature evidence is that depression is an important modulator of neurocognitive function [77]. In fact, the effect of depression on neurocognitive function was more important compared to COVID-19 severity or the presence of comorbidities up to 6 months post-infection [77,102].

### 4.3. Neurocognitive Deficits and SARS-CoV-2 Variants

Whether some SARS-CoV-2 variants, alpha, omicron or delta, confer a higher risk for neurocognitive impairment is inconclusive [77]. Based on the available evidence, an increased risk of epilepsy or seizures, neurocognitive deficits, insomnia, and anxiety disorders just after (versus just before) the emergence of the delta variant is proposed [92]. Albeit the limited to-date evidence, it is proposed that delta variant is more consistent with brain fog symptoms [77].

There is, also, limited to-date evidence on the potential effect of vaccination in reducing the severity of COVID-19 outcomes, including neurological sequelae [77,103]. Preliminary evidence suggests that potential vaccination prior to COVID-19 infection has no effect, although these results need to be treated with caution [103].

## 5. Development of Neurological Disease

The available data are consistent with an increased risk for the establishment COVID-19-induced systemic inflammation is prolonged and could promote neuroinflammation, thus making COVID-19 survivors more vulnerable for long-term neurological disorders in COVID-19 survivors [78,86,92,104] including neurodegenerative diseases (AD and PD) regardless of the course of the disease [41,42,105,106,107]. In addition to the risk for AD onset [31], case reports of acute parkinsonism onset following COVID19 infection in patients with no prodromal symptoms have raised concerns that COVID-19 pandemic can cause an increase in PD risk in the future [108,109,110,111,112]. COVID-19 survivors have a higher risk for the development of parkinsonism and dementia, compared to healthy controls or following adjustment for confounding factors, up to 12 months post-infection [78,86,92,104]. The risk can be potentially aggravated by factors that include older age (>60 years) and the severity of COVID-19 infection, although the risk was shown to be elevated even in people who did not require hospitalization during acute COVID-19 [104]. In the largest systematic review to date of 26,408 378 participants, pooled analysis demonstrated that COVID-19 conferred an increased risk of NOD (HR = 1.49, 95% CI: 1.33–1.68) in both genders regardless of age, which was sustained up to 2 years post-infection [79]. Following infiltration to the CNS, the interaction of SARS-CoV-2 with Toll-like receptor-2 may induce or accelerate neurodegenerative pathogenic mechanisms that can ultimately confer AD and PD [113,114]. In vitro experiments have also confirmed that SARS-CoV-2 can infect neurons leading to aberrant tau phosphorylation and neurodegeneration [115]. Interestingly, a Rhesus macaque model of SARS-CoV-2 neuroinfection produced both parkinsonism and Lewy body pathology [116].

However, it is inconclusive whether COVID-19 infection is a long-term risk factor for new-onset neurological diseases or whether it acts as a catalyst for pre-existing neurological conditions [78,86,92,104,117]. This is mainly due to the paucity of longitudinal studies, limitations associated with statistical methodology (confounder adjustment) and the availability of pre-COVID data, as well as the clinical and demographic characteristics of the participants.

## 6. Biomarkers of Neurocognitive Impairment in COVID-19

Elevated peripheral markers of neurodegeneration usually encountered in AD appear to characterize COVID-19 patients with cognitive impairment, adding to the overlap between COVID-19 and AD [28,32,74]. Complementing neurocognitive performance tests with evidence from robust indicators of neurocognitive impairment and hallmarks of AD can therefore be valuable. Compared to age- and sex-matched controls, LC patients with mild–moderate COVID-19 had inflammatory mediator levels 8 months post-infection, including IL-6 and INF-γ, that were consistent with immunological dysfunction [118]..

In the longitudinal case-control study by Duff et al. of pre- and post-pandemic assessments, COVID-19 patients had reduced plasma Aβ42:40 levels and poor cognitive performance in relevant assessments, which were predicted to be equivalent to 4 years of aging (based on an −0.5% change in the Aβ42:Aβ40 ratio per year of age) [32]. SARS-CoV-2 infection was associated with a statistically significant reduction in the Aβ42:40 levels (2% drop from pre-pandemic assessments), which was maintained even after adjustment for potential confounding factors. More vulnerable subjects, such as older and hospitalized subjects also had altered levels of other AD-related distinct biomarkers, such as increased p-tau-181 levels [32]. It was estimated that compared to matched controls, 75-year-old SARS-CoV-2 subjects had an extra 8.2% increase in p-ttau-181, and a decrease of 4.7% and 2.3% in Aβ42 and Aβ42:Aβ40 levels, respectively [32].

Increased levels in at least one inflammatory marker, compared to healthy controls or reference values, have been reported in LC patients with neurocognitive and neuropsychiatric deficits 12 or more weeks post-diagnosis [7,17,118]. These markers include proinflammatory cytokines, C-reactive peptide (CRP), D-dimer, and procalcitonin. In the meta-analysis by Ceban et al., 3.9–32.2% of LC individuals had increased IL-6 levels (3 pg/mL versus the reference value of ≤1.8 pg/mL), 1.8–24.5% had increased CRP levels (>2.9 mg/dL) and 9.8–38.0% had increased D-dimer levels [17]. Increased D-dimer levels are indicative of pulmonary embolism, thrombosis as well as severe virus infection. In COVID-19 patients, D-dimer levels over 0.5 mg/mL correlated with severe infection [119]. Higher maximum D-dimer levels, also, correlate with poorer functioning in verbal recall and psychomotor speed assessments [12,81]. Furthermore, statistically significant correlations between high ferritin levels or D-dimer levels with longer time for the completion of executive function assessments (Trail Making Test A and/or B), higher D-dimer and ferritin levels with poor performance in working memory assessments, and higher ferritin levels with poor performance in attention assessments have been reported [120]. Elevations in both CRP and IL-6 have, also, been reported in elderly (>62 years old) COVID-19 patients that presented with delirium as well as LC patients with mild–moderate disease course [9,64,96,118]. In one of the aforementioned studies, delirium emerged as an independent risk factor for neurocognitive decline in COVID-19 patients at hospital discharge [64]. Correlations between elevated CRP and slower reaction times in neurocognitive assessments or deficits in the sustained attention domain are reported as well [13]. Interestingly, increased D-dimer and CRP levels are significant negative prognostic factors that are associated with neurocognitive deficits, namely in verbal working memory, attention, perceptuomotor speed, mental flexibility, visuoconstructive skills, or visual memory, in patients with asymptomatic peripheral arterial disease (APAD) compared to matched controls [121]. APAD predisposes to cerebrovascular diseases and is highly prevalent among older people. This evidence highlights the potential for exploiting early neurocognitive impairment symptoms in COVID-19 patients as signs of increased risk of further complications.

The majority of studies investigating potential biomarkers of neurocognitive impairment in COVID-19 survivors are retrospective. A prospective observational longitudinal 12-month study following acute mild-to-moderate COVID-19 patients (n = 128) demonstrated that mild to moderate neurocognitive impairment, with symptoms that resembled brain fog, affected 16%, 23%, and 26% at 2, 4 and, 12 months, respectively, after diagnosis [122]. This study identified a statistically significant positive correlation between kynurenine pathway metabolites (quinolinic acid, 3-hydroxyanthranilic acid, and kynurenine) only, but not cytokines and chemokines (such IFN, IL-2/4/5/6/8/10, TNF-α, GMCSF) and neurocognitive impairment, which potentially warrants further investigation as a therapeutic target for COVID-19-related neurocognitive decline. A positive correlation between anosmia and neurocognitive decline at 2 months only was identified, and although anosmia was self-limiting and resolved over time; however, neurocognitive decline persisted [105].

## 7. Imaging Studies and Neurocognitive Impairment in COVID-19

Imaging assessments can complement and facilitate the understanding of the cognitive effects of COVID-19 [39,77]. Nevertheless, a limitation of neuroimaging studies is the lack of pre-infection imaging data to facilitate comparisons [77]. Significant loss of gray matter in addition to other areas of the brain is detected in imaging studies of COVID-19 patients, which were shown to persist up to five months post-infection [35,123]. Depending on the mechanism and area of the brain affected, the corresponding neurocognitive abilities are affected [123]. Functional imaging studies have demonstrated alterations in brain activity, particularly in the delta wave range, and functional connectivity changes in neurocognitively impaired COVID-19 survivors compared to healthy controls or pre-infection imaging data up to two months post-infection [18,124,125]. The alterations in functional connectivity between the caudate and the left precentral gyrus were related to the severity of the neurocognitive deficits, mainly in the executive functions of COVID-19 patients compared to age- and sex-matched controls as well as in the performance in dedicated neurocognitive tests [125].

Cechetti et al. performed a longitudinal evaluation of a small cohort of (n = 49) COVID-19 survivors with emergency room admission due to respiratory symptoms, of whom 86% were hospitalized, and presented with new-onset neurocognitive impairment with or without both hyposmia and dysgeusia versus healthy controls with no neurocognitive impairment at 1-month post-discharge (baseline) and 10 months later using both neurocognitive and imaging assessments (MRI and EEG), demonstarting that these assessments were inter-related [18]. In the aforementioned study, at baseline, 53% of subjects had impairment in at least one neurocognitive domain (namely 16% of subjects had only executive impairment and 6% of patients each had memory or visual-spatial impairment only, whereas 25% had a multi-domain impairment and no patient had language impairment), 28% had psychopathological disturbances (depression and post-traumatic stress disorder), 59% had dysgeusia and 45% had hyposmia. At the 10-month follow-up, 46% of subjects had impairment in at least one neurocognitive domain (namely 3% of subjects had only executive impairment, 6% of patients each had memory or visual-spatial impairment only, whereas 21% had multi-domain impairment) and 33% had psychopathological disturbances (depression and/or post-traumatic stress disorder). The reduction in neurocognitive impairment was accompanied by a reduction in delta band EEG connectivity, potentially due to improvements in executive impairment. Thus, EEG can be used for long-term neurocognitive assessments in COVID-19 survivor follow-up in clinical practice.

Other than EEG, changes in quantitative EEG (QEEG), which is acceptable as a supportive diagnostic tool in epilepsy, vascular diseases, dementia, and encephalopathy, were observed in COVID-19 patients with brain fog symptoms compared to pre-brain fog symptom onset [124]. QEEG reactivity has also been used to aid neurological recovery prognosis in critically ill hospitalized COVID-19 patients [126]. In addition to QEEG, 18F FDG PET scan is expected to assist in neurocognitive impairment assessments by corroborating structural information to clinical assessments [39,77,126]. Conversely, the combination of structural (EEG, MRI) and functional neuroimaging tools (functional MRI) can allow the integration of both structural and functional information to facilitate the understanding of LC-related neurocognitive impairment [36].

## 8. COVID-19 Neuroinflammation and Neurodegenerative Disease

Onset with a Focus on AD

The Alzheimer’s Association Global Consortium has proposed a framework to investigate the neurological complications of COVID-19 long-term and determine potential links between SARS-CoV-2 infection, brain function, neurocognitive symptoms, as well as AD and related dementias via an ongoing international longitudinal study that will include a combination of phenomenological description and neurocognitive and neurological assessments [13]. As AD affects mainly older adults, it was hypothesized that COVID-19 may evoke AD and related dementia by affecting the extended olfactory cortical network in older individuals with predisposing factors [13]. Bearing in mind that AD and COVID-19 share similar risk factors, it is possible that risk factors that may emerge following COVID-19, such as cerebrovascular disease and/or pre-existing risk factors, may converge following SARS-CoV-2 infection and trigger the development of AD [13].

Inflammoproteomics have identified inflammatory molecules that link AD and COVID-19. Aberrant type-I interferon signaling is also associated with neurocognitive impairment and neurodegeneration [58]. Inflammatory marker levels, including IL-6 and CRP, are down-regulated by adiponectin (ADPN) receptors, namely AdipoR1 and AdipoR2, that are expressed in various areas of the brain, including the hypothalamus, hippocampus, and cortex [72]. AdipoR1 is implicated in insulin sensitivity regulation, thus accounting for the contribution of diabetes as a risk factor for neurocognitive impairment. AdipoR2 promotes neural plasticity through the activation of biochemical pathways that inhibit inflammation and oxidative stress [72]. Adiponectin has anti-atherogenesis, anti-glycemic, and anti-inflammatory properties in addition to neuroprotective effects [72]. A case series demonstrated that ADPN levels were significantly decreased in the acute phase of COVID-19 in patients with respiratory failure compared to patients with non-COVID-19 respiratory failure [72].

The reported elevation of neurodegenerative biomarkers (e.g., glial fibrillary acidic protein GFAP, neurofilament light chain-NFL, tau proteins) or, conversely, the decreased levels of neuroprotective markers following COVID-19 infection lends further evidence to the connection between SARS-CoV-2 infection and neurodegenerative disease onset [105]. Various studies have demonstrated that the levels of biomarkers of neuronal injury, such as neurofilament light chain (NFL), correlate with encephalopathy, older age, severity of COVID-19 infection, and poor prognosis in hospitalized COVID-19 patients [28,34,44,105]. The levels of NFL, GFAP, and ubiquitin carboxy-terminal hydrolase L1 levels L1 (UCHL1) in hospitalized COVID-19 patients without a history of dementia were as high as in non-COVID AD controls [105]. Notably, in hospitalized COVID-19 patients with neurological complications, NFL and GFAP levels were higher complications or healthy controls [28]. Additionally, NFL levels inversely correlated with deficits in episodic memory, executive function, or overall global cognition in hospitalized severe COVID-19 patients up to 90–120 days following hospital discharge was logical [21,44]. Thus, NFL levels may be exploited not only for determining clinical outcomes but also for neurocognitive impairment prognosis, at least in hospitalized COVID-19 patients. Additionally, the levels of VEGF, which is a marker of endothelial damage, correlated with attention deficits in the aforementioned study, whereas the plasma concentration of proinflammatory chemokines and growth factors was found to be increased compared to the controls or non-infected patients with self-reported memory deficits (MCI group). The proinflammatory chemokine and growth factor profile of the COVID-19 patients was also persistently different compared to the MCI group [21].

Abnormalities in other factors, such as the neuropeptide orexin, which regulate sleep–wake states and are manifested with narcolepsy are also increasingly investigated in neurodegenerative disorders [127]. Although orexinergic abnormalities have been studied in PD and correlate with disease severity, they have been reported in other neurological diseases, including AD and correlate with Aβ pathology [128,129].

## 9. Conclusions

Neurological complications represent one of the most frequently reported extra-pulmonary manifestations of COVID-19. Neurocognitive deficits may develop secondary to COVID-19 and potentially persist long-term, regardless of disease severity. Additionally, neurocognitive impairment may trigger or exacerbate pre-existing mental or psychiatric comorbidities, which in turn can further aggravate neurocognitive impairment.

The available evidence presented here demonstrates that COVID-19 shares many similarities with AD at epidemiological, molecular, and imaging levels, as well as other neurodegenerative diseases such as PD. It remains to be clarified whether COVID-19 alters the natural history and onset of AD and related dementias or speeds their development. It is therefore suggested that COVID-19 may contribute to accelerated cognitive decline, though long-term and population-based studies are needed to establish its contribution globally. Older individuals as well as individuals with predisposing factors are at increased risk and can be complicated by confounding factors. A thorough assessment of patients’ medical history for potential confounders, complemented with neuroimaging and neurocognitive assessments across a wide range of COVID-19 severity levels, including asymptomatic cases, will contribute evidence on the potential link.

Clinicians should be aware of the link between COVID-19, neurocognitive impairment and neurodegenerative disease onset and be cautious about the potential emergence of neurodegenerative diseases in COVID-19 survivors, especially in high-risk patients. Additionally, healthcare decision-makers should also consider implementing preventive policies promptly to mitigate the long-term risks of LC on neurocognitive functioning overall. It is therefore imperative that LC patients, regardless of disease severity and self-reported neurocognitive deficits, are subject to long-term follow-up that includes regular neuroimaging, and neuropsychiatric and neurocognitive assessments. The frequency can be determined by the treating physicians, depending on the presence of risk factors for neurocognitive impairment as well as the severity of COVID-19. Attention and executive functions are the most frequently observed neurocognitive deficits, even from disease onset. The available cognitive assessment tests, such as Mini-Mental State Examination (MMSE) and Montreal cognitive assessment test, can be reliably used as a sensitive first-level approach for screening COVID-19 patients for global cognitive impairment [117]; these tools are also commonly used reliably for neurocognitive assessments in AD. Thus, these can be prioritized subsequently to global cognition assessments. Subsequently, second-level tests can be used to determine specific neurocognitive deficits [117].

The available evidence, to date, has also contributed information on potential biomarkers that could be exploited to complement neurocognitive and imaging assessments. Long-term follow-up of cognitively intact subjects who developed AD versus controls revealed changes in CSF biomarker levels—namely Aβ42, Aβ42:40, total tau, p-tau-181, NFL, and brain volume and cognitive performance that preceded AD onset by up to 20 years for Aβ42, while cognitive decline emerged six years prior to AD onset [30].

The magnitude of the cognitive sequelae of COVID-19 and impact on the epidemiology of neurodegenerative disease cannot be fully determined, to date, and will be revealed in the coming years. Careful planning and implementation of a post-pandemic strategy that will take into account a potentially increased incidence of cerebrovascular, cardiovascular, and neurodegenerative diseases in the coming years is required.

Cognitive decline is detrimental to employment, everyday activities, social functioning, and quality of life. The association between neurocognitive deficits and social cognition in other neurodegenerative diseases is well-established [130]. The extent to which neurocognitive deficits contribute to long-term disability in COVID-19 survivors remains to be seen in the coming years. However, their prompt detection is necessary for the implementation of effective rehabilitation strategies. COVID-19 survivors who have risk factors for AD should therefore be closely monitored and regularly evaluated over time.

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
