# Peer review of "Neurocognitive Impairment After COVID-19: Mechanisms, Phenotypes, and Links to Alzheimer’s Disease"

_brainsci, 2025, doi:10.3390/brainsci15060564_

Round 1

Reviewer 1 Report

Comments and Suggestions for Authors

Doskas et al present a review on the neurocognitive impairment after COVID. I have the following comments regarding the work:

  1. It would be valuable to refer to cognitive decline in two major neurodegenerative diseases in the context of COVID-19 - including Parkinson's disease
  2. Authors mention viral factors in neurodegeneration, perhaps the issue of HERV could be included.
  3. The issue of inflammation in neurodegeneration could be more extensively discussed - Ref. McGeer et al
  4. The context of COVID-19 as a factor accelerating cognitive deterioration should be further explored.
  5. Authors should consider acknowledging endocrine factors additionally impacting neurodegenerative diseases including cognitive deterioration as a possibly co-existing factor in the processes affected by COVID-19 - REF. Role of orexin in pathogenesis of neurodegenerative parkinsonisms. Neurol Neurochir Pol. 2023;57(4):335-343. doi:10.5603/PJNNS.a2023.0044

Author Response

Doskas et al present a review on the neurocognitive impairment after COVID. I have the following comments regarding the work:

Thank you for your valuable comments. The focus of the current publication was decided to be on links with Alzheimer’s disease. The aim of a future publication is to focus on neurological/neurodegenerative diseases in general.

  1. It would be valuable to refer to cognitive decline in two major neurodegenerative diseases in the context of COVID-19 - including Parkinson's disease The text has been re-arranged following feedback from reviewer 3 and the information on the risk of PD onset is more clear. Also, case reports of acute PD onset have been added to indicate that COVID-19 could be a risk factor for neurodegenerative disease onset altogether. We aim to release another publication that will focus on the incidence of neurological disease following COVID-19, including PD and AD, and provide further information there
  2. Authors mention viral factors in neurodegeneration, perhaps the issue of HERV could be included. It is added
  3. The issue of inflammation in neurodegeneration could be more extensively discussed - Ref. McGeer et al The reference is added and additional references, such as Zhang et al. 2023. The contribution of gut inflammation is also added
  4. The context of COVID-19 as a factor accelerating cognitive deterioration should be further explored.This is further reinforced in the conclusion. It is still early days to have conclusib=ve evidence because there are many confounding factors and we do not have yet long-term follow up studies
  5. Authors should consider acknowledging endocrine factors additionally impacting neurodegenerative diseases including cognitive deterioration as a possibly co-existing factor in the processes affected by COVID-19 - REF. Role of orexin in pathogenesis of neurodegenerative parkinsonisms. Neurol Neurochir Pol. 2023;57(4):335-343. doi:10.5603/PJNNS.a2023.0044 It is added

Reviewer 2 Report

Comments and Suggestions for Authors

Very comprehensive and up to date review. Tables summarizing findings would help the readers (for instance listing pathophysiological factors possibly at play).

Did the high mortality in older patients in the acute stage of COVID alter the epidemiology of dementia, at least for the short term?

 Is it fair to say that most of the world population has been exposed to COVID and this may be an issue in longitudinal studies comparing various populations? 

Author Response

Very comprehensive and up to date review. Tables summarizing findings would help the readers (for instance listing pathophysiological factors possibly at play). We have focused on meta-analyses in the table, therefore the patient backgrounds are quite heterogeneous to fit them in a table. Additionally demographic and clinical factors were investigated in the meta-analyses. We have highlighted the meta-analyses so that readers may be directed for further insights

Did the high mortality in older patients in the acute stage of COVID alter the epidemiology of dementia, at least for the short term? It is possible, because COVID-19 can lead to the premature death of individuals who were likely to develop dementia (eg people with established cognitive impairment, older adults) or fewer older adults may survive to ages that dementia manifests. This being said, it may act as a catalyst for the development of cognitive impairment that could lead to dementia in COVID-19 survivors, though the potential effect of protective/aggracating factors, have to considered. Therefore the effect on the long-term prevalence of dementia will reflect a balance of these factors.

Is it fair to say that most of the world population has been exposed to COVID and this may be an issue in longitudinal studies comparing various populations? It is highly likely and should be considered in clinical studies in relevant diseases, ie it should be investigated as a risk factor. Longitudinal studies would be the most useful source of information

Reviewer 3 Report

Comments and Suggestions for Authors

This manuscript by Doskas et al. is a critical review of literature on the relationship between COVID-19 disease caused by SARS-CoV-2 viral infection and neurological and neurodegenerative conditions that develop post-COVID-19 in many patients, with a focus on Alzheimer’s disease. The authors used systematic reviews and meta-analyses and a limited number of specific case studies or cohort studies choosing the most reliable ones and leaving out those lacking appropriate controls or those using too small sample. The question arises as to whether those reviews that have been utilized are not based on specific case studies or cohort studies which the current authors reject.

The manuscript is well written in terms of language but becomes more and more boring as one keeps reading. The main idea that long covid causes neurological and neurodegenerative problems such as a set of conditions and followed by percentages and the risk factors such as the severity of covid and age seem to be repeated again and again. Therefore the manuscript may benefit from a revision with an attempt to make the text more concise and less redundant.

Some specific comments:

Line 75: “New onset neurocognitive deficits…” Should it be “New onset of neurocognitive deficits…”

Lines 98-99: “lower Aβ42 and Αβ42:40 levels and increased p-Tau-98 181 levels are detected in the presymptomatic stage of AD.” Please clarify if the lower levels refer to the cerebral parenchyma, cerebrospinal fluid, blood plasma or else.

Lines 458-461:” COVID-19 patients had reduced plasma Aβ42:40 levels and poor cognitive performance in relevant assessments, which were predicted to be equivalent to 4 years of aging… SARS-CoV-2 infection was associated with a statistically significant reduction in the Aβ42:40 levels.” Please explain if plasma Aβ42:40 levels are increase or decrease in AD and how that level correlates with the Aβ42:40 ratio in the brain. What is meant by “equivalent to 4 years of aging?”

Author Response

This manuscript by Doskas et al. is a critical review of literature on the relationship between COVID-19 disease caused by SARS-CoV-2 viral infection and neurological and neurodegenerative conditions that develop post-COVID-19 in many patients, with a focus on Alzheimer’s disease. The authors used systematic reviews and meta-analyses and a limited number of specific case studies or cohort studies choosing the most reliable ones and leaving out those lacking appropriate controls or those using too small sample. The question arises as to whether those reviews that have been utilized are not based on specific case studies or cohort studies which the current authors reject. The systematic reviews and meta-analyses were selected to mitigate bias due to confounding factors and to allow more valid conclusions to be made. Cohort studies or case studies that contributed some information that could be of value were used if deemed appropriate. We did not screen the literature for all cohort studies, which were numerous

The manuscript is well written in terms of language but becomes more and more boring as one keeps reading. The main idea that long covid causes neurological and neurodegenerative problems such as a set of conditions and followed by percentages and the risk factors such as the severity of covid and age seem to be repeated again and again. Therefore the manuscript may benefit from a revision with an attempt to make the text more concise and less redundant. The text has been revised

Some specific comments:

Line 75: “New onset neurocognitive deficits…” Should it be “New onset of neurocognitive deficits…” It is amended

Lines 98-99: “lower Aβ42 and Αβ42:40 levels and increased p-Tau-98 181 levels are detected in the presymptomatic stage of AD.” Please clarify if the lower levels refer to the cerebral parenchyma, cerebrospinal fluid, blood plasma or else. The clarification is added in the text. I am quoting from the relevant references: Changes were seen approximately simultaneously for CSF and plasma biomarkers. Overall, plasma biomarkers had smaller dynamic ranges, except for CSF and plasma P-tau which were similar. In conclusion, using state-of-the-art biomarkers, we identified the first changes in Aβ, closely followed by soluble tau. 

Lines 458-461:” COVID-19 patients had reduced plasma Aβ42:40 levels and poor cognitive performance in relevant assessments, which were predicted to be equivalent to 4 years of aging… SARS-CoV-2 infection was associated with a statistically significant reduction in the Aβ42:40 levels.” Please explain if plasma Aβ42:40 levels are increase or decrease in AD and how that level correlates with the Aβ42:40 ratio in the brain. What is meant by “equivalent to 4 years of aging?” The clarification is added in the text

Round 2

Reviewer 1 Report

Comments and Suggestions for Authors

The manuscript was sufficiently improved.